# Acoustic Volume Rendering for
# Neural Impulse Response Fields

**Zitong Lan**[1]    **Chenhao Zheng**[2]    **Zhiwei Zheng**[1]    **Mingmin Zhao**[1]
[1]University of Pennsylvania    [2]University of Washington

## Abstract

Realistic audio synthesis that captures accurate acoustic phenomena is essential for creating immersive experiences in virtual and augmented reality. Synthesizing the sound received at any position relies on the estimation of impulse response (IR), which characterizes how sound propagates in one scene along different paths before arriving at the listener's position. In this paper, we present Acoustic Volume Rendering (AVR), a novel approach that adapts volume rendering techniques to model acoustic impulse responses. While volume rendering has been successful in modeling radiance fields for images and neural scene representations, IRs present unique challenges as time-series signals. To address these challenges, we introduce frequency-domain volume rendering and use spherical integration to fit the IR measurements. Our method constructs an impulse response field that inherently encodes wave propagation principles and achieves state-of-the-art performance in synthesizing impulse responses for novel poses. Experiments show that AVR surpasses current leading methods by a substantial margin. Additionally, we develop an acoustic simulation platform, AcoustiX, which provides more accurate and realistic IR simulations than existing simulators. Code for AVR and AcoustiX are available at `https://zitonglan.github.io/avr`.

## 1    Introduction

Our acoustic environment shapes every sound we hear – from the crisp echoes bouncing through hallways to the layered resonance of a symphony filling a concert hall. These spatial characteristics not only define our daily auditory experiences but also prove crucial for creating convincing virtual worlds [15, 60]. At the core of these spatial characteristics is the impulse response (IR), which captures the complex relationship between an emitted sound and what we hear. Like a unique acoustic fingerprint, the impulse response varies across different positions, encoding how sound waves interact with the environment through reflection, diffraction, and absorption [22, 41]. We can recreate the acoustic experience at any position by convolving the corresponding impulse response with any desired sound sources (e.g., music, speech). Given its foundational role in spatial audio synthesis, understanding and modeling the spatial variation of impulse responses in acoustic environments has emerged as a critical challenge and attracted increasing research attention [2, 26, 27, 29, 35, 36, 44, 49, 53].

Current approaches construct a neural impulse response field – a learned mapping that generates impulse responses given the emitter and listener poses. To model the high spatial variation of impulse responses, existing methods either fit a neural network to directly learn the field [29, 44] or rely on audio-visual correspondences to learn mappings from vision [26, 27]. While these methods can approximate the general energy trend, they struggle to capture the detailed characteristics of impulse responses, leading to incorrect spatial variation of impulse responses (Fig. 1).

We argue that a key barrier to achieving better performance is the absence of physical constraints that inherently enforce consistency across multiple poses. Without such physical constraints, the network

38th Conference on Neural Information Processing Systems (NeurIPS 2024).

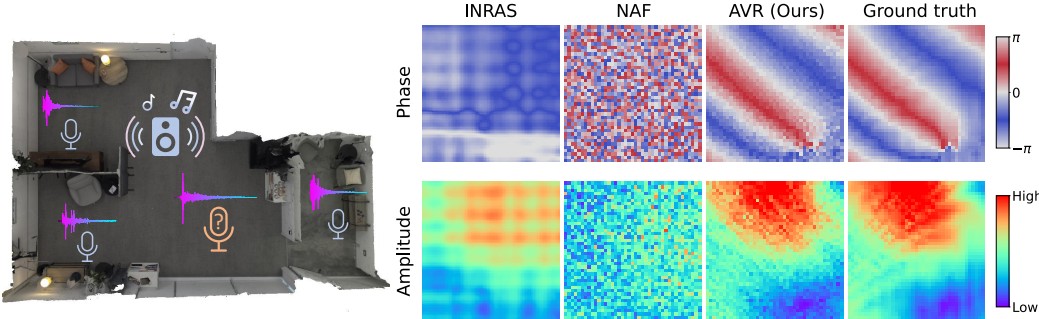

**Figure 1: Left:** From observations of the sound emitted by a speaker, our model constructs an impulse response field that can synthesize observations at novel listener positions. **Right:** Visualization of spatial variation of impulse responses on MeshRIR[20]. The synthesized impulse responses at different locations are transformed into the frequency domain, where we visualize phase and amplitude distributions at a specific wavelength (1m).

tends to overfit the training data and show poor generalizability. The received impulse response fundamentally arises from sound waves propagating through space, combining direct transmission with environmental reflections. This physical insight motivates us to develop a framework that inherently encodes wave propagation principles into the modeling of impulse response fields.

In this paper, we introduce Acoustic Volume Rendering (AVR) to model the field of acoustic impulse responses. Our approach draws inspiration from Neural Radiance Fields [33], which has demonstrated remarkable success in modeling 3D scenes by representing light transport through volume rendering. However, acoustic waves present several fundamental challenges that require adaptations to the volume rendering framework: First, acoustic impulse responses, unlike light transmission, are inherently time-series signals, with acoustic waves from different locations reaching the listener at varying delays. The issue is further compounded when dealing with discrete impulse responses sampled in the real world. Second, impulse responses exhibit high spatial variation, in contrast to images where neighboring pixels show strong correlations. This characteristic makes network optimization particularly challenging [43, 45]. Finally, unlike cameras that capture light with precise directional information (i.e., pixels), microphones capture combined signals from all directions.

To address these challenges, we convert impulse responses from the time domain to the frequency domain with Fourier transforms and perform volume rendering in the frequency domain. We apply phase shifts to the frequency-domain impulse responses to account for time delays, bypassing the limits of finite time domain sampling. The frequency-domain representation also exhibits lower spatial variation, facilitating network optimization. To account for signals from all possible directions, we cast rays uniformly across a sphere and use spherical integration to synthesize the impulse response measurements. Additionally, this design enables personalized audio experience by integrating individual head-related transfer functions (HRTFs) [57] into spherical integration at inference time. Our evaluation results show that AVR outperforms existing methods by a large margin in both simulated and real-world datasets [10, 20] and can zero-shot render binaural audio (Sec. 4.3).

In parallel with AVR, we develop AcoustiX, an acoustic simulation platform that generates more physically accurate impulse responses compared to existing simulators. While current simulators often introduce significant errors in signal phases and arrival times, AcoustiX produces impulse responses that better match the physical properties of real-world acoustics. Fig. 2 demonstrates the inaccuracies in impulse responses generated by SoundSpaces 2.0 [9]. Some existing simulators assign random phases when generating impulse responses [9, 40], which fails to reflect real-world acoustic behavior [4]. Since current research in impulse response synthesis heavily relies on simulated datasets [10, 29, 44], these simulation inaccuracies can impede progress in the field. To

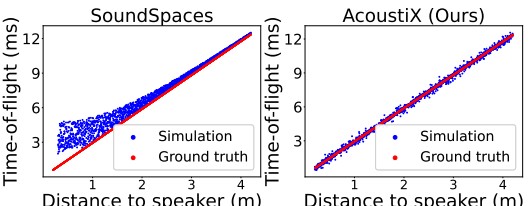

**Figure 2: AcoustiX for improved acoustic simulation.** Time-of-flight indicates how long it takes for an emitted sound to reach a listener. With sound traveling at a constant speed, the time-of-arrival should be proportional to the emitter-listener distance. While SoundSpace 2.0 simulations show significant time-of-flight errors, particularly at short emitter-listener distances, AcoustiX produces more accurate arrival times. All simulations are performed in the Gibson Montreal room [56] with direct line-of-sight between emitter and listener.

address these limitations, we develop a new simulation platform based on the Sionna ray tracing engine [16] and incorporate acoustic propagation equations to resolve the aforementioned issues. Similar to SoundSpaces 2.0, `AcoustiX` supports acoustic simulation in both user-provided 3D scenes and a variety of existing 3D scene datasets [24, 56].

In summary, this work makes the following contributions:

- We introduce acoustic volume rendering (`AVR`) for the neural impulse response field to inherently enforce acoustic multi-view consistency. We introduce a frequency-domain rendering method with spherical integration to address the challenges associated with acoustic impulse response modeling.

- We demonstrate that `AVR` outperforms existing methods by a large margin in both real and simulated datasets. `AVR` also supports zero-shot and personalized binaural audio synthesis.

- We develop `AcoustiX`, an open-source and physics-based impulse response simulator that provides accurate time delays and phase relationships through rigorous acoustic propagation modeling.

## 2 Related Work

**Impulse Response Modeling.** Traditional impulse response modeling [2, 32, 49] relies on audio encoding, which encodes collected data and spatially interpolates the impulse response for unseen positions [2, 32]. However, this approach comes with large memory costs [29] and struggles to generate impulse responses with high fidelity [10]. Machine learning techniques have been integrated in recent years to enhance the quality of synthesized impulse responses [35, 36, 37]. For instance, generative adversarial networks (GANs) have been utilized for more realistic acoustic synthesis [36, 37]. As implicit representations have become more popular, several works [26, 27, 29, 44] in recent years proposed neural implicit acoustic fields and achieved state-of-the-art performance. However, these learning-based methods still produce unsatisfying waveform shapes and show weakness in novel impulse response synthesis [10]. To this end, our learning-based approach further integrates wave propagation principles [22], synthesizing high-fidelity impulse responses.

**Neural Fields.** Since the success of NeRF [33], the concept of neural fields has been expanded comprehensively. Initially, incorporating depth supervision [12, 38, 51] into radiance fields proves helpful for novel view synthesis. Some following studies adopt occupancy [59] or signed distance functions [11, 54, 62], instead of density, to represent scenes. Later, integrating volume rendering with measurements from sensors of other modalities, such as time-of-flight sensors [3], LiDAR [18, 31, 55], and structured light imaging [42], has achieved great performance. Extensions [28, 61] have further modified volume rendering to model radio-frequency signals.

**Acoustic Simulation.** Acoustic simulation primarily relies on either wave-based or geometric approaches to approximate sound propagation in indoor environments. While wave-based methods [7, 14, 46, 47, 52] are generally more precise for low-frequency sound, they require significant computation for high-frequency signal simulation. For faster acoustic simulation, geometric approaches have gained considerable attention. These methods, such as image source [1, 5, 39] and ray tracing [8, 9, 21, 41, 46], are often used in practical applications like virtual reality. However, some commonly-used simulation platforms [8, 9] suffer from inaccuracies in time-of-flight calculation and phase simulation for sound propagation. `AcoustiX` ensures physics-based accuracy, facilitating further research on acoustic-related topics.

## 3 Method

Our objective is to learn an impulse response field for one scene to synthesize impulse responses for the unseen emitter and listener poses. The *impulse response* $h(t)$ quantifies the received signal at a specified listener location $p_l$ oriented by $\omega_l$, resulting from an impulse emitted from $p_e$ with orientation $\omega_e$. It encompasses how sound propagates within a specific scene. We begin by revisiting fundamental principles of acoustic wave propagation. We then introduce our impulse response field and our frequency-domain acoustic rendering. Lastly, we discuss the implementation specifics. In addition, we summarize the key features of our simulator `AcoustiX`.

## 3.1 Acoustic Wave Propagation Primer

We first explain the simplest form of acoustic propagation to highlight two basic properties of sound propagation: *time delay* and *energy decay*. Assuming an omnidirectional emitter at $p_e$ transmits a Dirac delta pulse $\delta(t)$ at time $t = 0$ uniformly into open space, the resulting signal at listener $p_l$ is given by [22]:

$$h(t) = \frac{1}{\|p_l - p_e\|_2} \delta(t - \tau), \quad \text{where } \tau = \frac{\|p_l - p_e\|_2}{v}, \tag{1}$$

and $v$ denotes the velocity of sound. The orientations of both the emitter $\omega_e$ and the listener $\omega_l$ are ignored under the omnidirectionality assumption. We note that the listener captures a *time-delayed* emitted signal, with an amplitude *decay* inversely proportional to the distance traveled. Additionally, the phenomenon can be alternately represented in the frequency domain with the Fourier Transform:

$$H(f) = \mathscr{F}\{h(t)\} = \int_{-\infty}^{\infty} h(t) e^{-j2\pi ft} dt = \frac{1}{\|p_l - p_e\|_2} e^{-j2\pi f\tau}. \tag{2}$$

With this representation, the time delay observed in the time domain manifests as a *phase shift* ($e^{-j2\pi f\tau}$) in the frequency domain.

## 3.2 Acoustic Volume Rendering

In a real scenario, sound emitted by a source undergoes complex interactions with the geometric structures of the environment. Each location in the scene may absorb some energy from the incoming wave, resulting in signal absorption; it may also reflect and scatter the wave, leading to signal re-transmission. To model these complex effects, AVR represents scene as a field: given an emitter location $p_e$ and its orientation $\omega_e$, the network $\mathbf{F}_\Theta$ outputs two key acoustic properties for any point $p$ in space given a direction $\omega$:

$$\mathbf{F}_\Theta : (p, \omega, p_e, \omega_e) \mapsto (\sigma, s(t)), \tag{3}$$

where $\sigma$ represents acoustic volume density and the time-varying parameter $s(t)$ models the acoustic signal transmitted out from the location $p$ in direction $-\omega$, including both initial emission and subsequent re-transmission.

With this parameterization, we now render the signal $h_\omega(t)$ received at a listener position $p$ from direction $\omega$, assuming the emitter is fixed and the impulse is emitted at time $t=0$. Similar to volume rendering for light, our process adopts volume rendering to accumulate the signals emitted from all locations along the ray $p(u) = p + u \cdot \omega$, with predefined near and far bounds $u_n$ and $u_f$. Differently, our approach also accounts for *time delay* and *energy decay* in acoustic signal propagation and performs alpha composition for time signals, resulting in our *acoustic volume rendering* equation:

$$h_\omega(t) = \frac{1}{tv} \int_{u_n}^{u_f} L(u)\sigma(p(u))s(t - \frac{u}{v}; p(u), \omega)du, \text{ where } L(x) = \exp\left(-\int_{u_n}^{x} \sigma(p(x))dx\right). \tag{4}$$

Note that each emitted signal $s(t - \frac{u}{v}; p(u), \omega)$ is associated with a time delay $\frac{u}{v}$. This delay accounts for the non-negligible sound propagation time, ensuring that the signal received by the listener at time $t$ originates from location $p(u)$ at the earlier time $t - \frac{u}{v}$. To account for *energy decay*, We apply a factor $\frac{1}{tv}$ to all the signals along the ray, independent of their emission time. Since all signals originate from the impulse transmitted at time 0, the traveled distance of any signal received at time $t$ is $tv$, whether it's from the original emission or a re-transmitted signal.

Listener receives signals from all directions, influenced by its gain pattern. The signal obtained from a single ray can not represent the whole received impulse response. To account for this, the final impulse response captured by the listener is a combination of signals from all directions:

$$h(t) = \int_\Omega G(\omega) h_\omega(t) d\omega, \tag{5}$$

where $G(\cdot)$ represents the listener gain pattern that characterizes the directivity of the listener, and $\Omega$ denotes the complete sphere of directions from which the listener receives signals. The whole acoustic rendering process is also shown in 3.

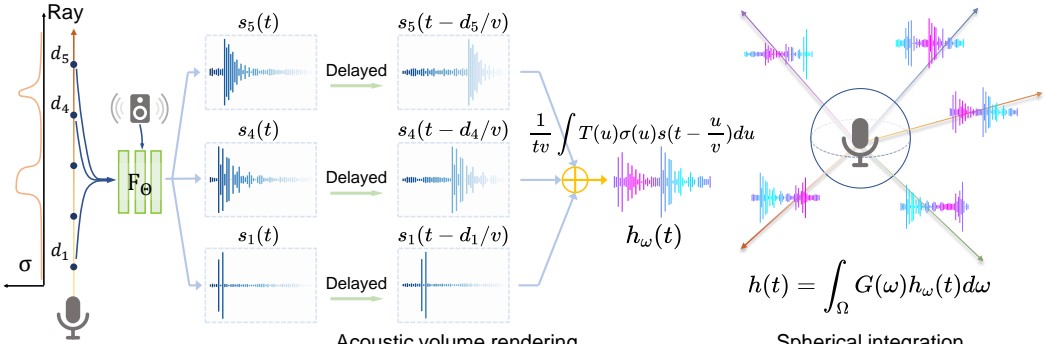

**Figure 3: Acoustic Rendering pipeline.** We sample points along the ray that is shot from the microphone and query the network to obtain signals $s(t)$ and density $\sigma$. Time delay $(\frac{d}{v})$ is applied to account for the wave propagation. After that, we combine signals and densities to perform acoustic volume rendering for each ray to get the directional signal ($h_{dir}(t)$). We integrate along the sphere to combine signals from all possible directions with gain pattern $G(\omega)$ to obtain the final rendered impulse response $h(t)$.

### 3.3 Acoustic Volume Rendering in the Frequency Domain

While the above method renders a continuous impulse response, the actual signal collected in the real world is discrete, sampled with a fixed interval $T$. This discretization converts the target impulse response from $h(t)$ to $h[n]$, where $h[n] = h(nT)$. Accordingly, we train neural networks to output $s[n]$ at these discrete timestamps. However, this discrete sampling presents a fundamental challenge in acoustic volume rendering. To capture the signal from one direction, $h_\omega[n]$, we need to evaluate the time-delayed signal $s(nT - \frac{u}{v})$. The key issue arises when $\frac{u}{v}$ is not a multiple of the sampling interval $T$, i.e., the required timestamps for delayed signals fall between our discrete samples, making accurate rendering difficult in the time domain. While one could add timestamp as an additional network input to interpolate between samples, this would require multiple network queries for each point, severely impacting rendering efficiency.

We address this challenge by reformulating the problem in the frequency domain. A key insight is that time delays in the signal correspond to phase shifts in the frequency domain (Eq. 2). This allows us to achieve arbitrary time delays $\frac{u}{v}$ by transforming the predicted signal $s[nT; p(u), \omega]$ into frequency domain $\mathscr{F}\{s[nT; p(u), \omega]\}$ and applying the corresponding phase shift, regardless of whether $\frac{u}{v}$ aligns with the sampling grid. Specifically, the delayed signal in the frequency domain can be obtained via:

$$\mathscr{F}\left\{s(nT - \frac{u}{v}; p(u), \omega)\right\} = \mathscr{F}\left\{s[nT; p(u), \omega]\right\} \cdot e^{-j2\pi fu/v}. \tag{6}$$

The linearity of the Fourier Transform $\mathscr{F}$ allows us to extend acoustic volume rendering to the frequency domain with the same alpha composition (i.e., integration) process:

$$H_\omega[f] = \mathscr{F}\left\{\frac{1}{tv}\right\} * \int_{u_n}^{u_f} L(u)\sigma(p(u))\mathscr{F}\left\{s(nT - \frac{u}{v}; p(u), \omega)\right\} du. \tag{7}$$

Here, the multiplication with energy decay factor $\frac{1}{tv}$ in Eq. 4 becomes the convolution with $\mathscr{F}\left\{\frac{1}{tv}\right\}$ in the frequency domain. Eq. 7 can be regarded as Eq. 4 in the frequency domain by applying discrete Fourier Transform on both sides of the equation. Similarly, the final impulse response in the frequency domain can be formulated analogously to Eq. 5:

$$H[f] = \int_\Omega G(\omega)H_\omega[f]d\omega. \tag{8}$$

Finally, time-domain impulse response $h[n]$ can be obtained through inverse Fourier Transform.

### 3.4 Sampling Rays and Points

To handle the integration in Eq. 7 and Eq. 8, our sampling strategy includes both ray sampling over a sphere and point sampling along a ray. We perform ray sampling by selecting $N_\theta$ azimuthal directions

and $N_\phi$ elevational directions, resulting in $N_\theta \times N_\phi$ distinct orientations that uniformly cover the sphere. For a ray along one of the directions, point sampling is conducted by evenly placing $N_r$ points between predefined near and far bounds, similar to [61]. Consequently, the network is queried at $N_\theta \times N_\phi \times N_r$ points for the synthesis of an impulse response. We refer readers to Appendix A for the details of our sampling strategy.

### 3.5 Optimization

We supervise the synthesized impulse responses $H[f]$ and $h[n]$ alongside the ground truth $H^*[f]$ and $h^*[n]$ in both frequency and time domains. We emphasize supervision in the frequency domain since the responses exhibit smaller local variability. In the frequency domain, $\mathcal{L}_{\text{spec}}$ measures spectral loss by comparing the real and imaginary components.

$$\mathcal{L}_{\text{spec}} = \|\text{Re}(H) - \text{Re}(H^*)\|_1 + \|\text{Im}(H) - \text{Im}(H^*)\|_1. \tag{9}$$

We also supervise the amplitude and phase of the synthesized signals with $\mathcal{L}_{\text{amp}}$ and $\mathcal{L}_{\text{phase}}$.

$$\mathcal{L}_{\text{amp}} = \||H| - |H^*|\|_1, \tag{10}$$

$$\mathcal{L}_{\text{phase}} = \|\cos(\angle H) - \cos(\angle H^*)\|_1 + \|\sin(\angle H) - \sin(\angle H^*)\|_1. \tag{11}$$

For time domain signals, $\mathcal{L}_{\text{time}}$ is employed to encourage amplitude consistency.

$$\mathcal{L}_{\text{time}} = \|h - h^*\|_1. \tag{12}$$

Our total loss is a linear combination of the above loss with different weights, including a multi-resolution STFT loss $\mathcal{L}_{\text{stft}}$ [58] and an energy loss $\mathcal{L}_{\text{energy}}$ similar in [30]:

$$\mathcal{L}_{\text{total}} = \mathcal{L}_{\text{spec}} + \lambda_{\text{amp}}\mathcal{L}_{\text{amp}} + \lambda_{\text{phase}}\mathcal{L}_{\text{phase}} + \lambda_{\text{time}}\mathcal{L}_{\text{time}} + \lambda_{\text{stft}}\mathcal{L}_{\text{stft}} + \lambda_{\text{energy}}\mathcal{L}_{\text{energy}}, \tag{13}$$

### 3.6 Simulation platform

`AcoustiX` uses Sionna ray tracing engine [16]. We modify the ray tracing in terms of ray interactions with the environment to support acoustic impulse response simulations. The simulator supports various ray interactions with the environment. Each material in the scene is assigned with frequency-dependent coefficients. This enables the tracing of cumulative frequency responses for each octave band to accurately simulate the impulse response. Room models can be created using Blender and exported as compatible XML files for our simulation setup. `AcoustiX` also supports the import of 3D room models from the iGibson dataset [24, 56]. More details can be found in Appendix D.

## 4 Experiments

**Implementation Details.** The input to our model is an emitter's pose (position $p_e \in \mathbb{R}^3$, direction $\omega_e \in \mathbb{R}^3$) and a 3D query point's pose ($p \in \mathbb{R}^3$, $\omega \in \mathbb{R}^3$), The model outputs the corresponding density $\sigma \in \mathbb{R}$ and discrete time signal $s[n] \in \mathbb{R}^\mathbb{T}$ at that query point. We first encode all input vectors into high-dimensional embeddings using hash grid [34]. The encoded input embeddings are then passed into a 6-layer MLP. The first 3 layers of MLP take as input locations $(p_e, p)$ and predict the density $\sigma$ and a 256-dimensional feature. The feature and encoded directions $(\omega_e, \omega)$ are then concatenated and passed into the last 3 layers, which outputs the signal sequence $s[n]$.

The sampling numbers used in the experiments are $N_\theta = 80$, $N_\phi = 40$, and $N_r = 64$. We set the weights of loss components to be $\lambda_{\text{amp}} = \lambda_{\text{phase}} = 0.5$, $\lambda_{\text{time}} = 100$, $\lambda_{\text{stft}} = 1$, $\lambda_{\text{energy}} = 5$. We train our model for 200 epochs for each scene. We use Adam optimizer with a cosine learning rate scheduler that starts at a learning rate $10^{-3}$ and decays to $10^{-4}$. The optimization process takes 24 hours on a single NVIDIA L40 GPU.

**Evaluation Metric.** We use comprehensive metrics to assess the quality of our method. Following [10, 29], we measure the energy decay trend by Clarity (C50), Early Decay Time (EDT), and Reverberation Time (T60). For the measurement of the correctness of waveform shape, prior works only consider the amplitude in the frequency domain (e.g. STFT error) to assess the performance, which we argue only indicates part of the waveform information: the transformed frequency signal is a complex number that is jointly defined by amplitude and phase. We therefore also include the frequency-domain *phase error* in our measurement: the L1 norm of the error in the cosine and sine

| Method | MeshRIR | | | | | | RAF-Furnished | | | | | | RAF-Empty | | | | | |
|---|---|---|---|---|---|---|---|---|---|---|---|---|---|---|---|---|---|---|
| | Phase | Amp. | Env. | T60 | C50 | EDT | Phase | Amp. | Env. | T60 | C50 | EDT | Phase | Amp. | Env. | T60 | C50 | EDT |
| AAC-nearest | 1.47 | 0.91 | 1.40 | 8.6 | 2.20 | 58.8 | 1.60 | 1.09 | 4.83 | 13.0 | 3.41 | 73.5 | 1.60 | 1.09 | 4.83 | 13.0 | 3.41 | 73.3 |
| AAC-linear | 1.44 | 0.89 | 1.42 | 8.2 | 2.29 | 58.9 | 1.60 | 0.99 | 3.81 | 12.4 | 3.65 | 90.2 | 1.59 | 1.10 | 5.22 | 13.1 | 3.25 | 71.5 |
| Opus-nearest | 1.45 | 0.72 | 1.37 | 5.2 | 1.26 | 35.7 | 1.60 | 1.19 | 5.35 | 14.4 | 3.78 | 80.3 | 1.59 | 1.16 | 4.58 | 13.3 | 4.25 | 100.6 |
| Opus-linear | 1.43 | 0.69 | 1.37 | 6.9 | 1.83 | 49.3 | 1.60 | 1.47 | 5.74 | 13.1 | 3.55 | 77.8 | 1.59 | 0.95 | 4.26 | 12.7 | 3.94 | 95.5 |
| NAF | 1.61 | 0.64 | 1.59 | 4.2 | 1.25 | 39.0 | 1.62 | 0.93 | 5.34 | 7.1 | 0.98 | 20.6 | 1.62 | 0.85 | 4.67 | 8.0 | 1.22 | 26.3 |
| INRAS | 1.61 | 0.77 | 1.85 | **3.4** | 1.47 | 40.7 | 1.62 | 0.96 | 6.43 | 6.9 | 1.08 | 21.4 | 1.62 | 0.88 | 4.72 | 7.6 | 1.21 | 25.8 |
| AVR (Ours) | **0.85** | **0.54** | **1.15** | 3.9 | **0.92** | 35.1 | **1.58** | **0.75** | **4.52** | **5.0** | **0.95** | **17.9** | **1.58** | **0.67** | **3.96** | **5.5** | **1.04** | **23.3** |

**Table 1: Quantitative results on real datasets (0.1s IR duration).** We report comprehensive metrics (lower is better) including phase error, amplitude error, envelop error(%), T60 reverberation time (%), clarity C50 (dB), and Early Decay Time (millisecond). AVR outperforms existing methods by a substantial margin. We note that the random phase error is 1.62, which means all learning-based methods except ours fail to learn valid phase information.

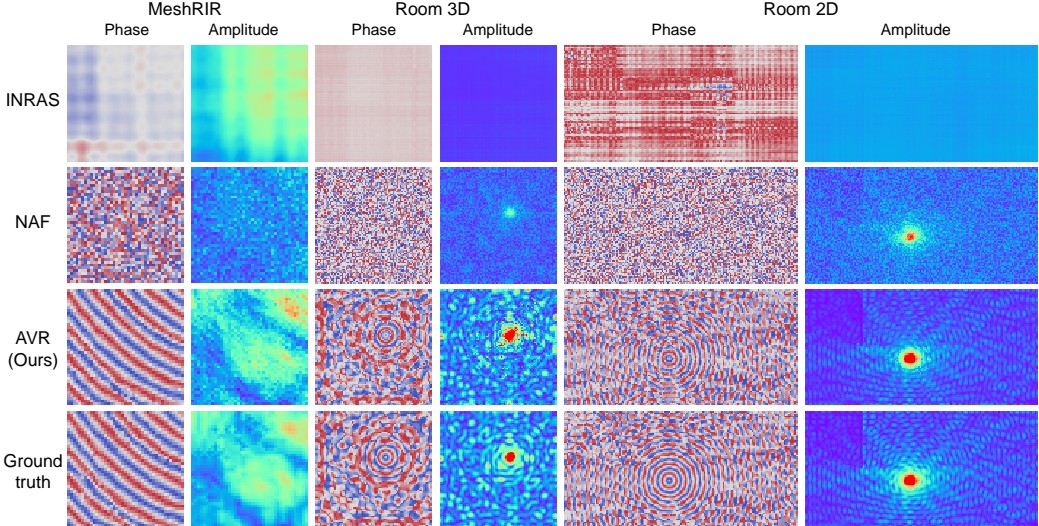

**Figure 4: Visualization of spatial signal distributions.** We compare the spatial signal distributions between ground truth and various methods on the MeshRIR dataset and two simulated environments. While NAF and INRAS fail to capture the signal distributions, our model can estimate amplitude and phase distributions accurately.

components of the phase. Besides the frequency domain, we also measure the time-domain impulse response signal accuracy by calculating the L1 error in the envelope, denoted as *envelop error*. Please refer to Appendix B for the detailed definition of each metric.

**Baselines.** We compare with both learning-based methods using neural implicit representations and traditional methods. NAF[29] is the first method that uses neural implicit representation to model the impulse response field. It uses an MLP to predict the spectrogram of the impulse response signals. Another method INRAS [44] disentangles the sound transmission process into three different learnable modules. Different from NAF, INRAS models the impulse response in the raw time domain. We also implement traditional audio encoding methods AAC [6] and Opus [50] and adopt the same setting as [29].

## 4.1 Results on Real World Datasets

We evaluate our model's performance on the datasets collected from real scenes. We adopt two commonly used room impulse response datasets: MeshRIR [20] and Real Acoustic Field [10]. MeshRIR collects monaural impulse response in a cuboidal room. We use S1-M3969 dataset split featuring a fixed single speaker for evaluation and the impulse responses are resampled to 24 KHz sampling rate. Real Acoustic Field (RAF) recorded monaural impulse responses in a real office space, with scenarios of the office being empty and the office being furnished. Different from MeshRIR, the speaker is directional and varies its position at different data points. The impulse responses in RAF are resampled to 16 KHz. All the impulse responses in these two datasets are cut to 0.1s. We use 90% of the data to train and the rest 10% for testing (Refer to Appendix C.2 for 0.32s on RAF dataset).

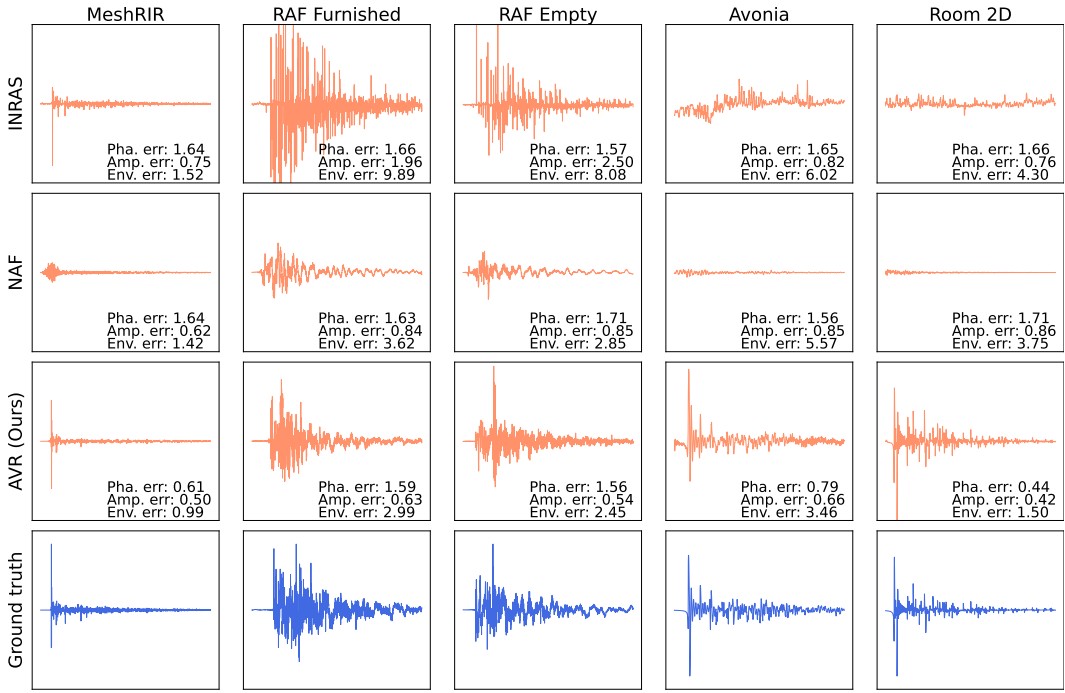

**Figure 6: Examples of synthesized impulse response with different methods.** We visualize the synthesized impulse responses as time signals (x-axis) on both real and simulated datasets. **Orange** lines represent model predictions and **Blue** lines represent ground truth. All plots share a common y-axis for easier comparison.

The results of all methods are shown in Tab. 1. We find that our method significantly outperforms the existing state-of-the-art baselines in all the datasets. We note that no existing learning-based method but ours performs better than chance in the phase error metric. In Fig.4 and Fig. 5, we show visualizations of the learned impulse response field for the entire scene. Our method captures much more spatial signal distribution than prior works. We also show an example of individual time-domain impulse response in Fig. 6. Although prior methods can capture the general decaying trend of the impulse responses, the waveforms (e.g. the peaks) are misaligned with the ground truth. In contrast, our method captures the waveforms much better. We especially point the readers to the time that the signal arrives for every impulse response, which indicates time-of-arrival. Our method has much smaller errors in terms of time-of-arrival due to our physics-based rendering.

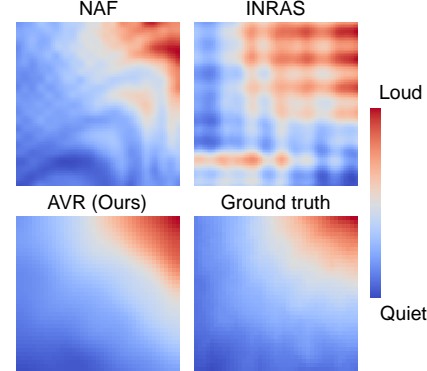

**Figure 5: Top-down view of loudness map on MeshRIR.** AVR predicts an accurate loudness map, while NAF and INRAS have inaccurate patterns.

## 4.2 Results on Simulation Dataset

Due to the high cost of equipment required to collect large-scale impulse responses in a real scene, MeshRIR and Real Acoustic Field are the only real-world impulse response datasets that are collected densely in an enclosed space. Researchers typically use simulated impulse response data in virtual 3D environments to complement the real-world datasets [26, 29, 44].

We use our simulation platform to simulate monaural impulse responses in three rooms and evaluate all methods' performance (Tab. 2). The simple 2D room is a 2D rectangular-shaped enclosed space with only one wall in the middle of the room, serving as a toy example. We also include two complicated 3D rooms from iGibson dataset [24, 56]. All rooms are equipped with a single omnidirectional speaker, with listeners placed randomly in the rooms. All the impulse responses are sampled at a 16 KHz sampling rate with 0.1s duration time. We follow the same split strategy used in real-world datasets.

| Method | 2D Room | | | | | | 3D scene Avonia | | | | | | 3D scene Montreal | | | | | |
|---|---|---|---|---|---|---|---|---|---|---|---|---|---|---|---|---|---|---|
| | Phase | Amp. | Env. | T60 | C50 | EDT | Phase | Amp. | Env. | T60 | C50 | EDT | Phase | Amp. | Env. | T60 | C50 | EDT |
| AAC-nearest | 1.40 | 0.88 | 2.88 | 11.0 | 2.77 | 32.6 | 1.62 | 0.95 | 6.52 | 10.4 | 2.01 | 27.3 | 1.62 | 0.95 | 5.57 | 10.4 | 2.01 | 27.9 |
| AAC-linear | 1.39 | 0.86 | 2.64 | 10.3 | 2.54 | 29.8 | 1.61 | 0.90 | 6.13 | 10.1 | 1.86 | 25.4 | 1.61 | 0.90 | 5.19 | 10.1 | 1.86 | 25.1 |
| Opus-nearest | 1.38 | 0.69 | 2.73 | 9.4 | 2.22 | 26.8 | 1.61 | 0.85 | 6.74 | 10.4 | 1.94 | 25.0 | 1.61 | 0.85 | 5.74 | 10.4 | 1.94 | 25.2 |
| Opus-linear | 1.34 | 0.67 | 2.53 | 9.1 | 2.21 | 25.6 | 1.70 | 0.71 | 6.31 | 9.8 | 1.87 | 24.0 | 1.60 | 0.71 | 5.32 | 9.8 | 1.87 | 24.3 |
| NAF | 1.62 | 0.69 | 6.72 | 7.6 | 2.02 | 23.5 | 1.62 | 0.99 | 9.12 | 10.2 | 2.12 | 25.4 | 1.62 | 0.81 | 5.53 | 7.9 | 1.38 | 19.1 |
| INRAS | 1.61 | 0.94 | 4.16 | 8.4 | 2.45 | 26.7 | 1.62 | 0.87 | 6.86 | **8.8** | 1.56 | 20.0 | 1.62 | 0.87 | 5.75 | **7.6** | 1.44 | 19.7 |
| AVR (Ours) | **1.04** | **0.65** | **2.35** | **7.5** | **1.62** | **20.5** | **1.52** | **0.63** | **5.93** | 8.9 | **1.44** | **19.1** | **1.53** | **0.65** | **5.16** | 7.6 | **1.23** | **17.1** |

**Table 2: Quantitative results on simulation dataset.** Our method significantly outperforms existing methods in both a 2D room with simple geometry and two 3D rooms with complex geometry.

Tab. 2 shows that our method consistently outperforms existing methods in all datasets. From the spatial signal distributions in Fig. 4, we observe that even in the simple 2D room the prior methods fail to accurately capture the accurate field distribution, while the field distribution generated by our method consistently matches the ground truth in both 2D and 3D cases. Our estimated time-domain signals at unseen poses are also closely matched with the ground truth signals, shown in Fig. 6.

### 4.3 Zero-Shot Binaural Audio Rendering.

AVR can generate accurate binaural audio despite being trained only on monaural audio modeling (without any fine-tuning). The existing method for rendering binaural audio either requires training at binaural channel spatial audio data [13] or manually creating signal delays. We render impulse response of left and right ears separately (20cm apart) in the MeshRIR scene. We play a piece of music 3 meters away from a listener, who turns its head from left to right and back again. We conduct a user study comparing the spatial perception of rendered binaural audio among NAF, INRAS, and our method. Seven users rated the similarity between expected head trajectories and their hearing experience on a 1-5 scale. Our method achieves the highest score of 4.71, compared to NAF's 1.42 and INRAS's 1.86. Other methods fail to synthesize accurate binaural audio because they are trained solely on monaural audio. Audio examples are available on our project website.

AVR is able to achieve binaural audio rendering for multiple reasons. First, our model captures accurate phase information in the impulse response to the extent that simply rendering the impulse response at the positions of the left and right ears can provide accurate phase differences, i.e., time delay or interaural time differences (ITD). Second, our model can easily incorporate the head-related transfer function for modeling the shadowing and pinna effects. Specifically, these direction-dependent filtering effects can be integrated into Eq.8 before summing responses from all directions. By replacing the direction-dependent weight term $G(\omega)$ with a direction-dependent HRTF function, we can achieve a more accurate binaural sound effect and reduce directional ambiguity (e.g., front versus back). Furthermore, explicit incorporation of HRTF allows our method to work with customizable HRTF for different users, allowing for an accurate and personalized listening experience.

### 4.4 Computing Efficiency

A comparison of runtime efficiency between AVR and other methods are shown in Tab. 3. This includes the inference time for different methods when they are trained to output IR of 0.1s and 0.32s. Since AVR uses acoustic volume rendering over a sphere, it is slower than the methods that directly output IR with a network. Encouragingly, various techniques have been proposed in recent years to significantly improve the efficiency of volume rendering

| Method | 0.1s IR | 0.32s IR |
|---|---|---|
| NAF | 3.2 ms | 6.4 ms |
| INRAS | 2.1 ms | 3.2 ms |
| AV-NeRF | 4.6 ms | 6.9 ms |
| AVR (Ours) | 30.3 ms | 90.7 ms |

**Table 3: Inference Time Comparison.**

and NeRF through efficient sampling strategies [17, 25, 48]. These approaches can be similarly adapted for acoustic volume rendering. More analysis on computing efficiency can be found in Appendix C.1.

### 4.5 Ablation Study

We ablate different choices of the sampling parameters during volume rendering, rendering domain, and loss components (Tab. 4). All models are evaluated on the MeshRIR dataset.

| Study Objectives | Variation | Phase. | Amp. | Env. | T60 | C50 | EDT |
|---|---|---|---|---|---|---|---|
| Sampling Parameters | 64 × 32 rays, 64 points | 0.956 | 0.547 | 1.17 | 4.07 | 1.20 | 47.6 |
| | 48 × 24 rays, 64 points | 1.356 | 0.607 | 1.37 | 4.57 | 1.73 | 67.6 |
| | 80 × 40 rays, 64 points | **0.847** | 0.535 | 1.15 | 3.86 | **0.92** | 35.1 |
| | 80 × 40 rays, 80 points | 0.857 | **0.529** | **1.14** | **3.79** | 0.95 | **34.9** |
| | 80 × 40 rays, 40 points | 0.869 | 0.543 | 1.17 | 4.66 | 1.30 | 52.9 |
| Rendering Domain | time-domain | 1.181 | 0.642 | 1.43 | 4.28 | 1.23 | 39.6 |
| | frequency-domain | **0.847** | **0.535** | **1.15** | **3.86** | **0.92** | **35.1** |
| Loss Component | w/o raw signal loss | **0.722** | 0.558 | 1.16 | 3.89 | 1.74 | 46.4 |
| | w/o angle & spec loss | 1.453 | 0.567 | 1.36 | 4.52 | 2.65 | 64.5 |
| | w/ all loss components | 0.847 | **0.535** | **1.15** | **3.86** | **0.92** | **35.1** |

**Table 4: Model ablations.** Performance for the model variants on MeshRIR dataset.

**Sampling Parameters.** We study the sensitivity of our model to sampling parameters $N_\theta$, $N_\phi$, $N_r$. We find that both increasing the ray numbers and the sampling points will both enhance the performance, but come with the cost of low training speed and high memory consumption.

**Rendering Domain.** We train our model using both time-domain volume rendering and frequency-domain volume rendering. Frequency-domain rendering effectively avoids issues associated with fractional time delays, aligning more accurately with the actual phenomenon of acoustic signal propagation. Consequently, this approach yields better results, confirming our argument in Sec. 3.3.

**Loss Component.** We also ablate loss components. We find that reducing any of the loss components results in decreased performance. However, it is noteworthy that all model variants, except for the one trained without the angle and spectral loss, outperform the baselines discussed in Sec. 4.1.

## 5 Discussion

**Limitations and Future Work.** Our rendering involves both spherically sampling rays and sampling points along each ray, which can lead to large memory consumption and longer inference time. Recently, many research works have been proposed to improve the efficiency of volume rendering and NeRF through efficient sampling strategies. We envision that similar methods could also be applied to acoustic volume rendering to speed up the rendering. Besides, AVR needs to train a new model for a novel scene, which requires effort to collect impulse response samples in the new scene. Future work could explore generalization to novel scenes by incorporating multi-modal inputs, aiming to synthesize an impulse response field using only a few visual or acoustic samples.

**Conclusion.** This paper proposes acoustic volume rendering to reconstruct impulse response fields that inherently encode wave propagation principles. We introduce frequency-domain signal rendering and spherical signal integration to address the unique challenges in impulse response modeling. Experimental results demonstrate that AVR significantly outperforms existing approaches. Additionally, we develop AcoustiX, an open-source simulation platform that provides accurate time-of-arrival measurements. Our work advances immersive auditory experiences in AR/VR, spatial audio in gaming and virtual environments, teleconferencing, and acoustic modeling in architectural design. Our realistic auditory simulations also benefit autonomous navigation, acoustic monitoring, and assistive hearing technologies where accurate acoustic modeling is essential.

## Acknowledgments

We thank the members of the WAVES Lab at the University of Pennsylvania for their valuable feedback. We are grateful to the anonymous reviewers for their insightful comments and suggestions.

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

# A  Sampling Rays and Points

The direction of a ray can be represented by two measures: azimuth $\theta$ and elevation $\phi$. We handle ray sampling by performing both azimuth and elevation sampling. For azimuth sampling, we apply stratified sampling between 0 and $2\pi$ to obtain another $N_\theta$ rays, where $i$ is the index:

$$\theta_i \sim \mathcal{U}\left[2\pi\frac{i-1}{N_\theta},\ 2\pi\frac{i}{N_\theta}\right]. \tag{14}$$

For elevation sampling, we evenly distribute $N_\phi$ rays, with $\phi_j = \arccos(2\frac{j}{N_\phi} - 1)$, where $j$ is the index. By combining all azimuth and elevation angles, we obtain $N_\theta \times N_\phi$ directions in 3D Cartesian coordinates, each represented as follows:

$$\omega_{ij} = [\cos\theta_i \sin\phi_j, \sin\theta_i \sin\phi_j, \cos\phi_j]. \tag{15}$$

For each sampled ray, we uniformly sample $N_r$ points on it. Given a ray $p(u) = p_l + u \cdot \omega$, starting from the point $p_l$ in direction $\omega$, the position of the $m^{th}$ point is given by:

$$p(u_m) = p_l + ((u_f - u_n)\frac{m}{N_r} + u_n)\omega. \tag{16}$$

With this sampling, we approximate the integral in Eq. 7 using quadrature as follows:

$$H_\omega[f] = \mathscr{F}\left\{\frac{1}{tv}\right\} * \sum_{m=1}^{N_r} T_m(1 - \exp(-\sigma_m\Delta u)\mathscr{F}\left\{s(nT - \frac{u}{v}; p(u_m), \omega)\right\},$$
$$\text{where } T_m = \exp\left(-\sum_{x=1}^{m-1}\sigma_x\Delta u\right), \text{and } \Delta u = \frac{u_f - u_n}{N_r}. \tag{17}$$

Combing our ray sampling strategy, we rewrite the Eq.8 and as follows:

$$H[f] = \sum_{i=1}^{N_\theta}\sum_{j=1}^{N_\phi} G(\omega_{ij})H_{\omega_{ij}}[f]. \tag{18}$$

# B  Evaluation Metric

***Envelope Error.*** Given the time domain ground truth impulse response $h^*[n]$ and our prediction $h[n]$, we can compute the envelope error by first obtaining the envelope using the Hilbert transform to get the analytic signal and then applying the absolute value, as follows:

$$\text{Env}^* = |\text{Hilbert}(h^*)| \tag{19}$$

The normalized *envelope error* is defined as follows (we multiply it by 100 to avoid small numbers):

$$\text{Envelope error} = 100 * \text{Mean}(\frac{|\text{Env}^* - \text{Env}|}{\max(\text{Env}^*)}) \tag{20}$$

***Phase and Amplitude Error.*** Given the frequency domain ground truth impulse response $H^*[f]$ and our prediction $H[f]$, we use a cosine and sine function encoded function to quantify the *phase error*:

$$\text{Phase error} = \text{Mean}(|\cos(\angle H^*) - \cos(\angle H)| + |\sin(\angle H^*) - \sin(\angle H)|). \tag{21}$$

The *amplitude error* is defined as follow:

$$\text{Amplitude error} = \text{Mean}(\frac{|abs(H^*) - abs(H)|}{abs(H^*)}). \tag{22}$$

# C  More Evaluation Results

## C.1  Computing Efficiency

We further examine the relationship between inference speed and the number of rays as well as the number of points sampled along each ray. As illustrated in Fig 7, the inference speed scales approximately linearly with both the number of rays and the number of points along each ray.

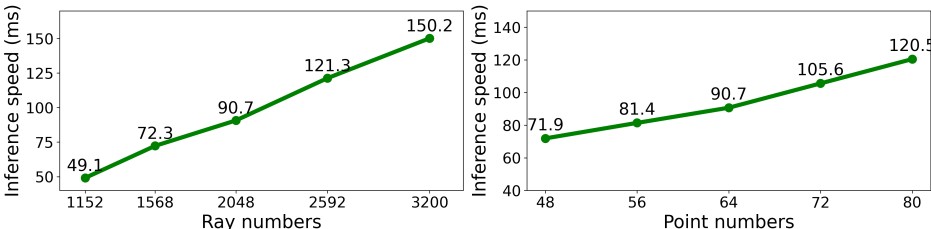

Figure 7: Impact of ray and point counts on inference speed.

## C.2 Additional Results on RAF Dataset

We repeated our experiments with a 0.32s RIR duration. Tab 5 shows the results on the RAF-Furnished dataset. We also included AV-NeRF as a baseline and multi-resolution STFT as an evaluation metric. With a 0.32s RIR duration, our method also outperforms these baselines. We provide loudness map visualization (Fig 8) for two different speaker positions at RAF-Furnished dataset with a grid size of 0.1m. Our method can better capture sound level differences caused by geometry occlusion and has a smoother spatial variation of loudness.

| Method | Phase | Amp. | Env. | T60 | C50 | EDT |
|--------|-------|------|------|-----|-----|-----|
| NAF | 1.62 | 0.79 | 1.67 | 7.68 | 0.64 | 24.2 |
| INRAS | 1.62 | 0.89 | 1.34 | 5.41 | 0.57 | 22.8 |
| AV-NeRF | 1.62 | 0.93 | 1.59 | 6.54 | 0.61 | 25.9 |
| AVR (Ours) | **1.59** | **0.69** | **1.04** | **4.95** | **0.55** | **19.8** |

Table 5: **Results on RAF dataset**. Performance comparison between our method and others on the RAF dataset with a 0.32s RIR duration.

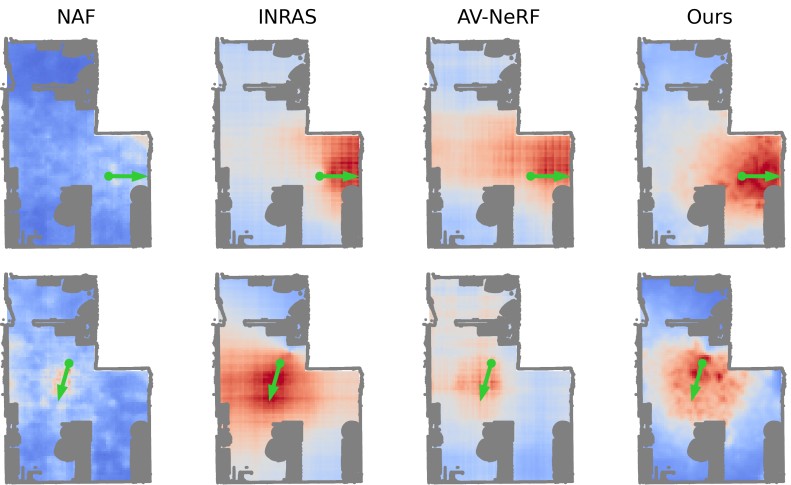

Figure 8: **Loudness map.** We visualize the loudness map of various methods using the RAF-Furnished dataset, which features the most complex structure among all the datasets we utilized. **Green** dots and arrows represent the speaker positions and orientations from a top view. **Gray** dots represent the room structures, outlining the geometry of walls, objects, and other elements.

## D Acoustic Simulation Platform

### D.1 Impulse Response Generation

`AcoustiX` is built based on Sionna [16] ray tracing engine that supports ray reflection, scattering, and diffraction. We modify the ray tracing engine in terms of ray interactions with the environment to support acoustic impulse response simulations. Each material in the scene is assigned a reflection coefficient $\beta$ and a scattering coefficient $\alpha$. For each reflection, the reflected wave's amplitude is

$E' = (1 - \alpha)\beta E$ with $E$ being the energy before the interaction. The scattered energy is $E'' = \alpha\beta E$. With these notations, the impulse response is formulated as follows:

$$h(t) = \sum_{n=1}^{N} \frac{A}{d_n} \delta_{\text{LP}} \left( t - \frac{d_n}{v} \right) \cdot \prod_{k=1}^{K_n} (1 - \alpha_{n,k})(-\beta_{n,k}), \tag{23}$$

where $d_n$ is the accumulated total length of $n^{th}$ path, $v$ is the velocity of sound in the air, $\alpha_{n,k}$ and $\beta_{n,k}$ denote material properties of $k^{th}$ reflection. We use the negative reflection coefficient definition discussed in [23]. In the equation, we assume purely specular reflection for simplicity. If scattering or diffraction occurs along the path, we replace the reflection term with the corresponding scattering or diffraction attenuation. $\delta_{\text{LP}}$ is a windowed sinc function defined similar to the one in [39].

In the implementation, we divide the whole frequency band into several octave bands and get their common acoustic properties. To assign frequency-dependent reflection and scattering coefficients to each material, we transfer Eq. 23 to the frequency domain and assign a coefficient to the amplitudes for each frequency band. The material coefficient is retrieved from [19].

## D.2 Acoustic Ray Tracing

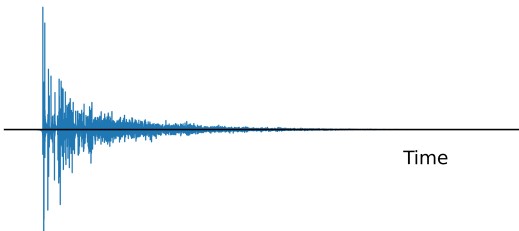

Time

**Figure 9: Example of a simulated impulse response**

In `AcoustiX`, users can determine the number of cast rays and maximum bouncing depth in the simulation. `AcoustiX` supports different ray-geometry interactions including reflection, scattering, and diffraction. By default, we enable all the functions above and set the maximum bouncing depth to 30 and the number of cast rays to 1e6 to enable comprehensive path searching within the rooms. We provide flexible API usage in `AcoustiX`, allowing users to adjust the acoustic ray tracing configurations and balance between simulation quality and speed. Fig. 9 shows an example of our simulated impulse responses.

## D.3 Room Model

`AcoustiX` supports customized room models. We create room structures with Blender, assign material names to all objects, and export the scene in XML formats using Mitsuba blender Add-on[1]. During simulation, each object is matched with its corresponding acoustic material properties by looking up a table mapping assigned names to properties. In addition to customizing room models, we also support importing 3D room models from the iGibson dataset [24, 56] into our simulations, assigning acoustic properties to each object.

# E Social Impact

As our method can synthesize high-quality impulse responses, our work can potentially enhance immersive VR/AR experiences and sound-dependent applications. `AcoustiX` fosters research and innovation in acoustic topics. Potential negative social impacts include the creation of misleading audio content, which could be used to deceive or manipulate users. For instance, high-quality impulse response generation could be exploited to fabricate realistic but fake acoustic environments or conversations, leading to misinformation or privacy violations.

---

[1]https://github.com/mitsuba-renderer/mitsuba-blender/tree/latest

