# OpenReview forum: "Acoustic Volume Rendering for Neural Impulse Response Fields"
_NeurIPS.cc/2024/Conference — NeurIPS 2024 spotlight_

### Official Review · Reviewer_RLmp · 2024-07-12

**Soundness:** 3
**Presentation:** 3
**Contribution:** 3
**Rating:** 6
**Confidence:** 4

**Summary:**

This paper addresses the challenge of novel sound synthesis from arbitrary positions. To achieve this, acoustic fields are modeled using an implicit neural representation, integrating acoustic wave propagation rules. Subsequently, a specifically designed volume rendering technique is employed to ensure consistency across diverse positions. Results from both simulated and real datasets validate the effectiveness of our approach. Additionally, the authors have developed a new impulse response simulation platform.

**Strengths:**

1. It is novel to employ volume rendering to ensure consistency across different positions, offering an advantage in capturing detailed waveform characteristics of impulse responses.
2. The proposed method demonstrates superior performance compared to state-of-the-art techniques.
3. The development of a new impulse response simulation platform is a valuable contribution.

**Weaknesses:**

1. The network is not very big (6 layers), I do not understand why the training takes 24 hours. Actually, the cited paper [29] is very fast, could the training time be reduced?
2. I enjoyed listening to the attached audio, which exhibits a strong sense of three-dimensionality. Including synthesized audios from other methods in the final Supplementary Material would enhance the comprehensiveness of the submission.

**Questions:**

See Weakness.

**Limitations:**

See Weakness.

---

> ### Author Rebuttal · Authors · 2024-08-07
>
> Thank you for valuable feedback and acknowledgement of our work. We answer questions point by point below.
>
> **W1: Network is not very big (6 layers). Why it takes 24 hours to train the model, while the Instant-NGP is fast? Could the training time be reduced?**
>
> Most of the runtime for our method is consumed by the acoustic volume rendering process, accounting for approximately 98% of the total, rather than by querying the network. This distinction from the typical visual NeRF setup results in longer training times compared to visual NeRF. Various techniques have been proposed in recent years to significantly improve the efficiency of volume rendering and NeRF through sampling strategies. We believe that these approaches can be similarly adopted and transferred for acoustic volume rendering, and we plan to explore this in our future work.
>
>
> **W2: Include synthesized audios from other methods in the final supplementary material.**
>
> Thanks for your feedback.  We also render binaural audios for other methods (NAF and INRAS) and will include it in the camera-ready version through link.

---

> > ### Comment · Reviewer_RLmp · 2024-08-12
> >
> > Thank you for your response. All of my concerns have been addressed.

---

> > > ### Author Response · Authors · 2024-08-13
> > >
> > > We are grateful for your acknowledgement that all of your concerns have been addressed. Thank you for your time and effort in reviewing our work.

---

### Official Review · Reviewer_DqiU · 2024-07-12

**Soundness:** 3
**Presentation:** 3
**Contribution:** 4
**Rating:** 7
**Confidence:** 4

**Summary:**

This method proposes a physics-based approach for learning neural impulse response field through implicit wave propagation modeling. This is the first work that really incorporates acoustic wave propagation principles for constructing an impulse response field, leveraging volume rendering techinques in analagy to how it's applied in NeRF. There is also a new acoustic simulation platform introduced for simulating more realistic impulse respones. Experiments show that the proposed method outperforms several previous learning based methods.

**Strengths:**

- There are many so-called audio-NeRF in prior work, but none of them actually are. This work is the most physics-based neural impulse response method I have seen, and the closest to what an audio-NeRF should look like.

- The problem is well-formulated, and it's an elegant formulation to implicitly model wave propagation, and use volume rendering for synthesizing room impulse responses.

- There are very informative visualizations and ablation studies that shows the advantage of the proposed method, especially on modeling the phase of the signal.

- The new simulation platform introduced will also be very useful for the community, and the comparison with SoundSpaces suggests that the community should start to use more realistic simulation platforms.

- Experiments on both simulated and real datasets have demonstrated the proposed method outperforms a series of prior baselines.

- Generally, the paper is also very well written with sufficent details and illustrations.

**Weaknesses:**

- The paper introduces a new acoustic simulation platform, which is claimed as one major contribution of the paper. However, in the main text, there is no detailed description of the simulator, and only says it is based on Sionna [16]. I then found it in Supp, but it would be good to also briefly summarize the key characteristics of this new simulator in the main text.

- Since it's the first method that implicity models wave propagation for neural impulse response rendering, it would be important to also compare with prior physics-based methods that is geometry-based to highlight the superity of modeling wave propogation, apart from those learning based methods. For example, how would the proposed method compare to this recent work https://arxiv.org/abs/2406.07532, that models the acoustic field using an image-source differentiably method.

- It would also be good to have some separate comparisons for high-frequency components vs. low-frequency components, and far-field vs. near-field to better understand the limitations and strength of the proposed framework. For example, for equation 1, the 1 / d assumption might not hold well for near field.

- The qualitative examples are somewhat lacking. It would be much more informative to include some qualitative comparisons to prior work.

**Questions:**

Apart from the questions in the Weakness section, another question I have is related to the zero-shot binaural audio experiment. The paper claims that the proposed method can even render binaural audio even without HRTFs, which is a bit confusing to me. Even if the simulator can render binarual audio by synthesizing the sound heard at the left and right ears, there are other ear/head-related effect are not factored in, so how can binarual audio be directly generated?

**Limitations:**

There are some limitations discussed, but it would be great to hear from the authors on more discussions on limitations especially on the compute efficiency (e.g., how much time it takes for the network to converge?) and memory usage compared to prior work, so that it can give peope the complete picture on the pros and cons of the proposed framework.

---

> ### Author Rebuttal · Authors · 2024-08-07
>
> Thank you for a thoughtful review, valuable feedback and acknowledgement of our work. We provide point by point clarifications and answer questions below.
>
> **W1: Summarize the key characteristics of the simulator in the main text.**
>
> Thank you for this suggestion, we will expand the description of simulation setup in the main text:
>
> Our acoustic simulator uses Sionna ray tracing engine. We modify the engine in terms of ray interactions with the environment to support acoustic impulse response simulations.
> The simulator supports various ray interactions with environment. Each material in the scene is assigned with frequency-dependent coefficients. This enables the tracing of cumulative frequency responses for each octave band to accurately simulate the impulse response.
> Room models can be created using Blender and exported as compatible XML files for our simulation setup. Our engine also support the import of 3D room models from the Gibson dataset.
>
> **W2: Comparison with physics-based methods.**
>
> The method in the paper (https://arxiv.org/abs/2406.07532) needs a precise measurement of the simplified room geometries and approximate them as a series of planes. While these parameters are provided by the author in the original paper's dataset, we do not have these measurements in our datasets to guarantee reproducible results. We leave the comparison with image-based method for future work.
>
>
> **W3.1: Separate comparisons for high-frequency components and low-frequency components.**
>
> In the RAF-Furnished dataset, we conducted additional experiments using the multi-resolution STFT loss to analyze both high-frequency (above 2 kHz) and low-frequency (below 2 kHz) components. The resulting errors are 0.91 and 1.07, respectively.
>
> **W3.2: Limitation in applying 1/d to near field.**
>
> In the near field, the acoustic field is difficult to explicitly formulate. We use 1/d to approximate the free space loss in paper, and allow our model to learn complex interactions through each point's signal and density.
>
>
> **W4: Lack of qualitative examples with prior works.**
>
> In the paper, we provide qualitative results on time domain impulse response and frequency domain signal distribution. Additionally, we provide loudness map visualization for two different speaker positions at RAF-Furnished dataset with grid size 0.1m (Please refer to supplementary PDF Figure 1). Our method can better capture sound level differences caused by geometry occlusion and has smoother spatial variation of loudness. We would also include qualitative results on our binaural sound rendering.
>
> **Q1: Confusion about binaural audio rendering**
>
> Please refer to **Zero-shot binaural audio rendering** part in the global rebuttal.
>
> **Computing efficiency:**
>
> Please refer to **Computing efficiency** in the global rebuttal.

---

> > ### Comment · Reviewer_DqiU · 2024-08-12
> >
> > Thank the authors for the additional comparisons and clarifications. The efficiency comparison is especially helpful for readers to fully understand the tradeoff of the proposed method, and I would encourage the authors to include a brief version of it in the main paper. This is a very interesting paper, and I would be very happy to see it appear in the conference and also see the open-sourcing of the new simulator for the community to use. Therefore, I am keeping my original rating and enthusiastically recommend the acceptance of the paper.

---

> > > ### Author Response · Authors · 2024-08-13
> > >
> > > We appreciate your suggestion and will incorporate the efficiency comparison into the main paper. We are also excited about the opportunity to open-source our simulator and contribute to the community. Thank you for your enthusiasm!

---

### Official Review · Reviewer_WhmA · 2024-07-13

**Soundness:** 1
**Presentation:** 2
**Contribution:** 3
**Rating:** 7
**Confidence:** 4

**Summary:**

This work presents a method to render a neural impulse response field to generate room impulse responses at novel positions after being trained with a number of limited RIR samples. The method works similar to a Nerf, rays are cast from a sphere and sampled at discrete positions along the rays. Here the signal at each sample point is transferred into the frequency domain and a phase shift is applied to delay them in time. The goal is to render the acoustic signal transmitted and volume density for any observer-emitter pair with known orientations and locations.

The approach uses among other methods two datasets to compare itself against common baselines in the domain. The authors emphasize having the best phase reconstruction compared to other results.

**Strengths:**

The method has great potential to correctly learn and reconstruct the phase in synthesized RIRs. It uses physical properties of wave-propagation in a very explicit way, compared to black box approaches, which can also help interpretability of the results.

The work potentially provides an improved simulator and, if certain points are improved, a good alternative to current simulators.

The authors remark very interesting issues with SoundSpaces, such as the time-of-flight error. They also found a very intuitive illustration for the sound pressure and phase field in a room which can again help with interpretation of results and comparison with other approaches.

The figures are very clear and the method is clearly described apart from notable omissions (see weaknesses and questions).

**Weaknesses:**

There are several odd observations and design decisions in this paper which at least should be explained. Their impact on performance is potentially great as can be probably seen in their baseline comparison. They differ by a large amount from the existing literature. Also, runtime (inference time) and impact of design decisions on it are not discussed but probably a major hurdle for this method. Without further explanation about the comparison with the state of the art and the validity of the method this contribution can not be sufficiently judged:

Especially since one of the arguments is about issues with sampling, the relation to the sampling rate used should not go uncommented. The authors decide to resample impulse responses to 16 kHz yet the RAF dataset is sampled at 48 kHz. Does that mean in Table 1 all other methods are also compared at 16 kHz? In the RAF paper, Table 7 shows results for this 16 kHz case for NAF and INRAS. They combine the furnished and empty case but for example, Inras has a T60 error of 5.34. In Table 1 of this paper, Inras is described as having errors of 11.3 and 10.6 in the furnished and empty case. Why is the baseline roughly twice as bad as reported in that paper? The EDT values of INRAS seem to have an opposite relation with 0.019 sec (RAF paper) vs. 57.4 MICROseconds and 62.9 MICROseconds. Either the authors mean milliseconds, in that case the results are worse without explanation, or they actually mean microseconds, in that case the results of Inras are way better in the RAF paper also without explaining why. Most of the other results are also not consistent with the RAF paper. The author need to explain why their baseline results seem to be so much worse than in the dataset paper.

If the explanation is the different split the authors use, they should explain why they did not use the split of 80%, 5%, 15% for training validation and test as used in the RAF paper or why a difference of 5% in the split leads to these vastly different results.

Another reason could be cutting the samples to 0.1 s. RAF compares samples at 0.32 s.

Large churches may have over 10 seconds of RT60. A typical living room may have 0.5 s RT60. Limiting the method to 0.1 s seems arbitrary.

The supplementary material plays music which sounds as if it starts towards the front left and then moves behind the head. Figure 7 shows a different route of the music. I did on purpose listen first to the audio before I read the description because expectation can change this experience. It is possible to imagine the sound being in the front or the back and both being plausible. This is only my experience and a number of test-listeners would be necessary to verify my findings. However, for me the audio experience is different than real binaural recordings which make me wonder if using a HRTF can really be omitted. If the authors simply create a time-delayed signal but without shadowing effects of the head and other important factors of the HRTF this is not binaural sound, this is stereo and then the directional ambiguity in perception can be explained. The authors should go into detail why they consider this zero-shot binaural sound.

This is a minor point and I does not reflect at all on the quality of the manuscript or the research but I would like to suggest to the authors to rename N_azi and N_ele to N_az and N_el to be consistent with other notations such as N_pt and avoid the distraction which could stem from reading the variable name N_azi wrong.

Little mistakes:
- In equation 12 the font of L_time is inconsistent with line 205
- Line 503 has one additional "frequency" word
- Line 509 starts with IIn

**Questions:**

- Why is 0.1 s chosen as duration time?

- What is the inference time of this method?

- What is the expected computation time if a duration of 1 second would be used?

- How is the inference speed dependent on the number of rays? Optimization seems to take 24 hours for 200 epochs which is not bad but if this can only be used for very short RIRs in very small rooms or with a reduced number of rays these are important limitations which should be discussed or measured and explained.

- How do the authors choose the size of the sphere they sample on? For example is it room dependent? Do they need the measurements of the room? What about complex scenes like apartments with connected rooms that are not shoebox like?

- Why do the authors not compare to AV-NeRF in table 1. The results from the RAF paper could have been taken to compare against a visual-audio method if split, sampling etc. would have been the same as in the RAF paper.

- What is the intuition/reson to concatenate directions only into the last 3 layers of the MLP?

- What is the bouncing depth of 30? Is that meters? Is that because 0.1 * speed of sound makes bouncing beyond 30 m irrelevant?

- Why is the phase important for this use case? The authors argue with VR applications. Many audio and audio-visual tasks ignore phase. If the method, after correcting the issues mentioned with the evaluation, ends up being closer to the other methods or even worse, then what is the benefit of reconstructing phase?

- To compare truly with the RAF paper, what is the STFT error? Having the waveform the authors could calculate the multi-resolution STFT loss to compare over the whole dataset instead of comparing waveforms on limited examples in figure 6.

**Limitations:**

The biggest limitation coming from using a duration of 0.1 seconds seems unexplained.

Inference time and impact of more complex rooms or longer durations seems important but is unexplained.

---

> ### Author Rebuttal · Authors · 2024-08-07
>
> Thank you sincerely for the thoughtful review and valuable feedback. Below, we address the questions raised and outline the revisions that we will include in the camera-ready version of the manuscript.
>
> **W1: Binaural sound rendering.**
>
> Please refer to **Zero-shot binaural audio rendering** in the global rebuttal.
>
> **Q1: Why is 0.1s chosen as RIR duration?**
>
> Thank you for raising this question! We chose to use 0.1s since the vast majority of the reflected energy shows within the first 0.1s. Specifically, with the RAF dataset, the average energy density within the initial 0.1s is around 500 times larger than that between 0.1s and 0.32s. The difference between 0.1s used in our evaluation and the 0.32s used in RAF caused the difference in evaluation metrics. (The microsecond was indeed a typo, it should be **millisecond**).
>
> To provide a better comparison with the results reported in RAF, we repeated our experiments with a 0.32s RIR duration. Table 1 shows the results on the RAF-Furnished dataset. We also included AV-NeRF as a baseline and STFT as an evaluation metric based on your suggestions. With a 0.32s RIR duration, the performance of the baseline methods aligns closely with the numbers reported in the RAF paper, and our method continues to outperform these baselines. We will include results for both 0.1s and 0.32s in the camera-ready version if accepted.
>
>
>
> | Method | STFT | Phase | Amp. | Env. | T60 | C50 | EDT |
> |:-------------:|:--------------:|:--------------:|:-------------:|:-------------:|:-------------:|:-------------:|:-------------:|
> | NAF       | 1.54        | 1.62             | 0.79             | 1.67             | 7.68            | 0.64                 | 0.024                |
> | INRAS     | 1.41        | 1.62             | 0.89                  | 1.34                  | 5.41                 | 0.57                 | 0.022                |
> | AV-NeRF                          | 1.49                  | 1.62                   | 0.93                  | 1.59                  | 6.54                 | 0.61                 | 0.025                |
> | **Ours**                    | **1.28**         | **1.58**          | **0.69**         | **1.05**         | **4.85**        | **0.51**        | **0.019**       |
> ||
>
>
> **Q2, Q3, Q4: What is the inference time of the method? What is the inference time if 1 second impulse response is to be rendered? How is the inference speed dependent on the number of rays?**
>
> Please refer to **Computing efficiency** in the global rebuttal for a detailed discussion on the runtime performance.
>
>
> **Q5: How to choose the size of the sphere? Is it room dependent? Does it need the measurements of the room? What about complex scenes like apartments with connected rooms that are not shoebox like?**
>
> We empirically choose the radius of the sphere (i.e., furthest point to sample on each ray) based on the rough size of the scene. This is the same as visual NeRF which needs a far point for sampling on each ray. Our radius parameter does not require precise measurements or detailed room models, unlike methods such as INRAS.
>
> **Q7: What is the intuition to concatenate directions only into last 3 layers of the MLP?**
>
> The intuition is that density only depends on position, while signal depends on both position and direction. In our design, the density prediction only take position as input. The last 3 layers  predict directional-dependent signal, so they will additionally take directions as input. Many visual volume rendering works share a similar network design.
>
>
> **Q8: What is the bouncing depth of 30?**
>
> Bouncing depth in ray tracing refers to the number of times a ray of light or sound can reflect off surfaces before being terminated.
> For higher-order bouncing events, the acoustic ray's energy diminishes to a level that can typically be neglected. To balance the computation efficiency and precision, we choose 30 as our maximum bouncing depth in the simulation setup.
>
> **Q9: Why phase is important?**
>
> Phase in the frequency domain is crucial for modeling RIR since time delay is equivalent to a phase shift in the frequency domain. Not using or being able to benefit from phase information might be a missed opportunity in prior work and could partially explain their underperformance.
>
> For binaural perception in VR applications, accurate sound localization is vital for creating immersive environments. Phase differences between the left and right ear are essential for the human auditory system to determine the direction of sound sources. Incorrect phase information can lead to errors in perceived direction and distance, breaking the illusion of a virtual space.

---

> ### Comment · Reviewer_WhmA · 2024-08-10
>
> I would like to thank the authors for their detailed explanations, additional experiments and clarifications. My biggest concerns about comparison with existing work, scalability and being explicit about the computation, have been well addressed.
>
> Given the potential positive impact in the domain, providing phase, and due to the excellent work in providing further analysis I have changed my recommendation to accept.

---

> > ### Author Response · Authors · 2024-08-13
> >
> > We are pleased to hear that your concerns regarding comparisons with existing work, scalability, and computation details have been resolved satisfactorily. Thank you for your support and recognition of our work!

---

### Official Review · Reviewer_4897 · 2024-07-13

**Soundness:** 3
**Presentation:** 4
**Contribution:** 4
**Rating:** 8
**Confidence:** 3

**Summary:**

The authors reformulate neural fields to model the spatial interactions of sound, by learning to predict impulse response in a fixed transmitter / arbitrary receiver pose geometry. The method substantially outperforms contemporary work on simulated and real world datasets, and lays the groundwork for new immersive AR/VR applications.

**Strengths:**

1. The methodology is well-motivated and creative in its incorporation of acoustic / wave propagation principles in the learning algorithm

2. Performance improvements over related work are nontrivial, in both simulated and real datasets

3. The proposed method stands out in its ability to recover the phase of synthesized waves, which is crucial for many time-critical applications

4. Frequency-domain rendering is a clever way to address the challenges in fractional time delay / discrete sampling in predicting IPR

5. A new simulation platform (when released) would be a significant enabler to other researchers

**Weaknesses:**

1. I feel that the ablation study + associated commentary is limited (see my questions in the next section). This is exacerbated by the fact that performance metrics are reported without standard deviations. I saw that the authors' rationale in the checklist for omitting error bars was budget, but I still think that expanding this section to include experiments on different network sizes, loss coefficients, etc. would increase this work's utility to researchers with different computational budgets, and better elucidate the impact of each component of the method. It would be nice to expand the commentary in the discussion with the authors' hypothesized explanations for results as well.

2. The paper lacks a crucial discussion on the computational complexity and runtime of the proposed method compared to existing approaches.

**Questions:**

1. How does the computational complexity of your method compare to existing approaches, particularly for real-time applications?

2. In lines 282-294 (discussion of zero-shot binaural audio rendering), are there any user studies / qualitative evaluations that could characterize the zero-shot performance vs. a network optimized for that particular task?

3. In table 3, it seems counterintuitive that removing raw signal loss improves phase error.. do the authors have any insight into this result?

4. In lines 302-304, its not clear to me why frequency-domain rendering gives significantly better results than time-domain rendering.. I understood the claim that the former is more compute-efficient, but don't see the claim that this should occur in Section 3.3 as the authors state -- could the authors clarify this?

**Limitations:**

The authors do acknowledge memory consumption (due to omnidirectional ray sampling) as a limitation. It would be good to expand on other potential limitations like generalization across environments, or applicability to time-critical applications.

---

> ### Author Rebuttal · Authors · 2024-08-07
>
> Thank you for a thoughtful review and valuable feedback. We address the questions related to our work below.
>
> **Q1: Computing efficiency and runtime of framework.**
>
> Please refer to **Computing efficiency** in the global rebuttal.
>
> **Q2: Include user study of qualitative evaluations about the zero-shot performances of binaural sound rendering.**
>
> Please refer to **Zero-shot binaural audio rendering** in the global rebuttal.
>
> **Q3: Ablation study explanations.**
>
> 1. Why does removing the time-domain loss improve the phase error?
>
> We use both time-domain and frequency-domain losses to supervise our training. When the time-domain loss is removed, the frequency-domain loss becomes the primary focus, causing the network to prioritize matching the frequency-domain signal over capturing time-domain characteristics. This leads to improved phase accuracy, as the network is more focused on aligning frequency components. However, this also means the network may fail to capture certain time-domain signal variations, resulting in poor performance in metrics related to temporal accuracy, such as C50 and EDT.
>
> 2. Why does frequency domain rendering give better result?
>
> As the reviewer pointed out, frequency-domain rendering effectively avoids issues associated with fractional time delays. This is because time delay representation in the frequency domain is continuous, aligning more accurately with the actual phenomenon of acoustic signal propagation. Consequently, this approach yields better results.
>
> 3. Ablation study on network size.
>
> We change our network layer from 5 to 8 to expand the ablation study in Table 3 of the paper. We demonstrate that increasing the network beyond six layers does not result in significant performance variations.
>
>
> | Network layer | Phase.         | Amp.           | Env.          | T60           | C50           | EDT           |
> |:-------------:|:--------------:|:--------------:|:-------------:|:-------------:|:-------------:|:-------------:|
> | 5             | 0.901          | 0.575          | 1.17          | 4.12          | 0.98          | 36.8          |
> | 6             | **0.847** | 0.535          | **1.15** | 3.86          | **0.92** | **35.1** |
> | 7             | 0.855          | 0.548          | 1.16          | 3.97          | **0.92** | 35.7          |
> | 8             | 0.849          | **0.533** | 1.16          | **3.82** | 0.93          | 35.9          |
>
>
> **Other Potential Limitations.**
> Thank you for this suggestion! We will expand the limitations to include efficiency, and generalization across environments.

---

### Author Rebuttal · Authors · 2024-08-07

We would like to express our gratitude to all the reviewers for their insightful comments and feedback. Below, we address two common questions raised by multiple reviewers: computing efficiency and zero-shot binaural audio rendering. The rest of the questions and comments are addressed individually for each reviewer.

**Computing efficiency**

To provide a more comprehensive understanding of our model's performance, we have conducted an additional analysis of its runtime efficiency and compared it with other methods. The results and analysis provided below will be incorporated into the camera-ready version of the paper if accepted.

We first compare the inference time of our model and other methods in Table 1 below. This includes the inference time for different methods when they are trained to output RIR of 0.1s and 0.32s.
Not surprisingly, our proposed method that uses volume rendering based on physical properties of wave propagation is significantly slower than the methods that directly output RIR with a network.
Encouragingly, various techniques have been proposed in recent years to significantly improve the efficiency of volume rendering and NeRF through efficient sampling strategies [1,2]. We believe that these approaches can be similarly adopted for acoustic volume rendering, and we plan to explore this in our future work.


|         | 0.1s RIR | 0.32s RIR |
|:-------:|:--------:|:---------:|
| NAF     | 3.2 ms   | 6.4 ms    |
| INRAS   | **2.1 ms**  | **3.2 ms**    |
| AV-NeRF | 4.6 ms   | 6.9 ms    |
| **Ours**    | 30.3 ms  | 90.7 ms   |
||


We also analyze how the number of rays and the number of points along each ray would impact the performance and inference time in Table 2.
Our method becomes more accurate with increased amount of sampling for rays and points.
At the same time, reducing the number of rays and points can proportionally reduce the inference time, without significant performance loss.
When it comes to real-time applications, besides all the potential approaches that can be employed to enhance the runtime efficiency of our method, predictive algorithms can be used to pre-calculate potential impulse responses based on likely user motion, which helps reduce perceived latency.

| Variation                | STFT | Phase | Amp. | Env. | T60  | C50  | EDT   | Runtime  |
|:---------------------------------:|:----:|:-----:|:----:|:----:|:----:|:----:|:-----:|:--------:|
| 80x40 rays x 64 pts | 1.17 | 1.58  | 0.65 | 1.03 | 4.79 | 0.48 | 0.018 | 150.2 ms |
| 72x36 rays x 64 pts | 1.21 | 1.58  | 0.68 | 1.03 | 4.83 | 0.50 | 0.019 | 121.3 ms |
| 64x32 rays x 64 pts | 1.28 | 1.58  | 0.69 | 1.05 | 4.85 | 0.51 | 0.019 | 90.7 ms  |
| 56x28 rays x 64 pts | 1.35 | 1.58  | 0.70 | 1.08 | 5.46 | 0.53 | 0.023 | 72.3 ms  |
| 48x24 rays x 64 pts | 1.45 | 1.59  | 0.69 | 1.06 | 6.34 | 0.61 | 0.026 | 49.1 ms  |
||
| 64x32 rays x 80 pts | 1.15 | 1.58  | 0.67 | 1.02 | 4.81 | 0.48 | 0.019 | 120.5 ms |
| 64x32 rays x 72 pts | 1.19 | 1.58  | 0.68 | 1.02 | 4.84 | 0.51 | 0.019 | 105.6 ms |
| 64x32 rays x 64 pts | 1.28 | 1.58  | 0.69 | 1.05 | 4.85 | 0.51 | 0.019 | 90.7 ms  |
| 64x32 rays x 56 pts | 1.33 | 1.59  | 0.69 | 1.07 | 4.93 | 0.50 | 0.020 | 81.4 ms  |
| 64x32 rays x 48 pts | 1.31 | 1.59  | 0.71 | 1.09 | 5.05 | 0.52 | 0.020 | 71.9 ms  |
||


[1] Turki, Haithem, Vasu Agrawal, Samuel Rota Bulò, Lorenzo Porzi, Peter Kontschieder, Deva Ramanan, Michael Zollhöfer, and Christian Richardt. "HybridNeRF: Efficient Neural Rendering via Adaptive Volumetric Surfaces." In Proceedings of the IEEE/CVF Conference on Computer Vision and Pattern Recognition, pp. 19647-19656. 2024.

[2] Hu, Tao, Shu Liu, Yilun Chen, Tiancheng Shen, and Jiaya Jia. "Efficientnerf efficient neural radiance fields." In Proceedings of the IEEE/CVF Conference on Computer Vision and Pattern Recognition, pp. 12902-12911. 2022.

**Zero-shot binaural audio rendering:**

Zero-shot in our case refers to the fact that our model can generate accurate binaural audio despite being trained only on monaural audio modeling (without any fine-tuning). Our method is able to achieve this for multiple reasons. First, our model captures accurate phase information in the RIR to the extent that simply rendering the impulse response at the positions of the left and right ears can provide accurate phase differences, i.e., time delay or interaural time differences (ITD).
Second, our model can easily incorporate the head-related transfer function (HRTF) for modeling the shadowing and pinna effects. Specifically, these direction-dependent filtering effects can be integrated into Equation 8 before summing responses from all directions. By replacing the direction-dependent weight term with a direction-dependent HRTF function, we can achieve a more accurate binaural sound effect and reduce directional ambiguity (e.g., front versus back).
Furthermore, explicit incorporation of HRTF allows our method to work with different HRTF from different users, allowing for accurate and personalized listening experience.

The rendered binaural audio provided in the supplementary material demonstrates accurate modeling of ITD by simply rendering at the left and right ears. We will include in the camera-ready version the binaural audio samples with the baseline methods for qualitative comparison. We note that **no** HRTF function was applied for our binaural rendering, and therefore there was directional ambiguity (e.g., front vs back). We will also include an audio sample with HRTF incorporated with our method.

---

> ### Author Response · Authors · 2024-08-08
>
> Continue of **Zero-shot binaural audio rendering**:
>
> We conducted an additional user study to compare the spatial perception of rendered binaural audio among NAF, INRAS, and our method. Seven users listened to each audio sample and evaluated the similarity between the expected head trajectories and their hearing experience. They then rated the similarity on a scale of 1 to 5, with 5 being the highest similarity and 1 being the lowest. Results show that our method achieved the highest similarity score of 4.71, while NAF and INRAS scored only 1.42 and 1.86, respectively. We will add this additional user study in the camera-ready version if accepted.

---

### Decision · Program_Chairs · 2024-09-25

**Decision:**

Accept (spotlight)

**Comment:**

The paper received uniformly positive reviews. Reviewers highlighted its potential impact on the physics-based neural impulse response field (RLmp, DqiU, WhmA, 4897), the quality of results (RLmp, DqiU, 4897), the novelty of the approach (RLmp, DqiU, 4897), and the clear presentation (WhmA). While there were initial concerns regarding computational complexity, missing comparisons, and a lack of important details, these issues were effectively addressed in the authors’ rebuttal.

After thoroughly reviewing the paper, the reviews, and the authors’ rebuttal, the AC concurs with the reviewers’ highly positive consensus and therefore recommends the paper for acceptance (spotlight).

For the camera-ready version, the authors should ensure all discussions from the rebuttal are incorporated into the main paper and supplementary materials. The specific changes that need to be implemented are:

**1. Improved presentation**: Summarize the key characteristics of the simulator in the main text (DqiU)

**2. Additional experiments**: Include an efficiency comparison (DqiU), results with a 0.32s RIR duration (WhmA), and user-study results for zero-shot binaural audio rendering.

**3. Audio results**: Include synthesized audio examples from other methods (RLmp).